# SNAP: Stopping Catastrophic Forgetting in Hebbian Learning with Sigmoidal Neuronal Adaptive Plasticity

## Abstract

Artificial Neural Networks (ANNs) suffer from catastrophic forgetting, where learning on new tasks causes the degradation of performance on previous ones. Existing algorithms typically use linear weight updates, where the magnitude of the update is independent of the current weight strength. This contrasts with biological neurons, which at intermediate strengths are very plastic, but consolidate with Long-Term Potentiation (LTP) once they reach a certain strength. We hypothesize that this biological mechanism could mitigate catastrophic forgetting in ANNs. We introduce Sigmoidal Neuronal Adaptive Plasticity (SNAP), an artificial approximation to Long-Term Potentiation for ANNs by having the weights follow a sigmoidal growth behavior, allowing the weights to consolidate and stabilize when they reach sufficiently extreme values. We compare SNAP to linear and exponential weight growth and see that SNAP prevents the forgetting of previous tasks for Hebbian Learning but not for Stochastic Gradient Descent (SGD) based learning.

## 1 Introduction

Continual learning is a remarkable human ability that allows for the sequential acquisition of new tasks while minimizing disruptions to previously learned knowledge. This capability supports the accumulation of skills and information throughout one's lifetime. This learning paradigm differs from the traditional offline training methods of ANN's, which assume a static dataset that is fully available at train time. As such, ANNs struggle with the process of continual learning, often suffering from catastrophic forgetting, where new learning significantly degrades performance on earlier ones (Parisi et al., 2019).

Traditional deep learning approaches assumes training data is independent and identically distributed (i.i.d.), which clashes with the sequential nature of continual learning (French, 1999). Even in humans, training on i.i.d. data can lead to suboptimal performance, suggesting that conventional models may not effectively replicate the learning dynamics observed in biological systems (Carvalho & Goldstone, 2014; Flesch et al., 2018).

### 1.1 Hebbian Learning

ANNs are primarily learned through backpropagation (Rumelhart et al., 1986), a method that requires knowledge of downstream signals of a given network layer. While this has been shown to be an effective algorithm for a number of downstream tasks, there is as of yet no evidence that a biological mechanism akin to backpropagation exists in the brain (Lillicrap et al., 2020). Additionally, the supervised nature of traditional backpropagation methods necessitates a large amount of pre-labeled training data, which are costly to generate.

Hebbian learning is a biologically plausible learning method that is described by the adage "neurons that fire together wire together." Concretely, it is a plasticity rule that is only dependent on the pre and post-synaptics responses of a neuron. The simplest formulation of this rule is expressed with the a local weight update algorithm given by

$$\Delta W_{ij} = \alpha x_j y_i \tag{1}$$

for input $x_j$, output $y_i$, and learning rate $\alpha$ (Kuriscak et al., 2015). Further refinements to this rule include Oja's rule (Oja, 1982) and Sanger's rule (Sanger, 1989), also known as the Generalized Hebbian Algorithm:

$$\Delta W_{ij} = \alpha y_j x_i - y_j (\sum_{k=1}^{i} W_{kj} y_k) \tag{2}$$

which has applications in principle component analysis. While multiple Hebbian learning rules exist, the general idea of a Hebbian algorithm is to use a learning rule that is spatially and temporally local, that is, the only information that is required to update the weights for a given layer of the network is the response of that layer to a given input signal. Due to this, Hebbian networks avoid the locking of layers that are waiting for top-down feedback, making them more amenable to parallel computation, as well as decreasing memory overhead (Li et al., 2025).

## 1.2 SNAP

Hebbian Learning and SGD trained models employ what we will call linear weight growth, where the magnitude of the weight update is independent of the current magnitude of the weights or the amount of updates already done (cf. Fig. 1a). In contrast, the brain's learning process includes distinct phases: initial rapid learning, where new information is quickly acquired but also easily forgotten, and Long-Term Potentiation (LTP), where synaptic connections stabilize. During LTP, learning does not strengthen further, but the acquired knowledge becomes resistant to forgetting. showcasing a non-linear pattern of neuroplasticity.

Inspired by these biological insights, we propose Sigmoidal Neuronal Adaptive Plasticity (SNAP) a novel approach to weight growth by imitating the brain's learning and consolidation phases. This method captures the essence of LTP, where synaptic weights initially grow rapidly but eventually stabilize, maintaining learned information without further increases in strength. We explore two variants of SNAP. In the first, synapse-wise SNAP (s-SNAP), plasticity is defined at the level of each individual weight and depends only on the value of the weight. Thus, in s-SNAP, a neuron can have weights with different levels of plasticity. In the second, neuron-wise SNAP (n-SNAP), plasticity is defined at the level of the neuron and depends on all the input weights. Thus in n-SNAP all the input weights of a neuron have the same plasticity. In that case, the plasticity of the neuron is controlled by the norm of the vector of input weights to the neuron.

We test out different weight growth behaviour: linear, sigmoidal, and exponential. We test both neuron-wise and synapse-wise plasticity, and both Hebbian and SGD based learning. We find that in the i.i.d. condition all types of weight growth behaviour can achieve good performance. In the sequential learning condition however, sigmoidal weight growth prevents catastrophic forgetting in Hebbian Learning but not in SGD based learning where it only slightly reduces catastrophic forgetting. To the best of our knowledge, this is the first time that catastrophic forgetting has been solved in Hebbian Learning without making use of replay (or pseudo-replay) mechanisms.

## 1.3 Contributions

1. We propose SNAP, a novel weight growth algorithm inspired by Long Term Potentiation in biological brains.

2. We define two versions of SNAP, namely s-SNAP and n-SNAP, which implement SNAP at the synapse and neuron levels respectively.

3. We implement SNAP with networks that are learned using both Hebbian and SGD based algorithms. These networks are then trained on classification datasets that are trained and compared on both in the i.i.d setting and in the continual learning setting.

## 2 Related Works

### 2.1 Long Term Potentiation

Long term potentiation, refers to a persistent increase in the strength of hippocampal synapses due to short, high-frequency excitations (Nicoll, 2017). Long term depression (LTD) refers to the converse effect. First introduced in 1973 (Bliss & Lømo, 1973), it has gone on to be one of the most attractive models of cellular memory. Various computational models of the mechanisms that govern synaptical strength and memory have been proposed. An early model of artificial memory is the Hopfield network (Hopfield, 1982), which stores memories as a set of $n$ binary vectors $\zeta_1, ... \zeta_n$ in a network. Inputs to the network $x$ can be thought of as perturbed memories $x = \zeta_k + \delta$ for some index $k$ (Tyulmankov, 2025). The network will then evolve according to a dynamics equation

$$x_i^{(t+1)} = sign(\sum_{j=1}^{N} w_{ij} x_j^{(t)}) \tag{3}$$

If the input $x$ truly correponds to an incomplete memory, the fixed point of this system will be the memory $\zeta_k$ that $x$ was perturbed from. This process can be thought of as a form of "pattern completion", and similar processes have been shown to exist in the CA3 region of the hippocampus, which exhibits LTP among recurrent synapses (Rolls, 2013).

Migliore et al. (1995) propose a model that considers a presynaptic signal $I$ that evenutally produces a retrograde $K$. This is reminiscent of backpropagation, with the noticeable difference that the "error" signal determined solely by spatially local variables. The strength of this retrograde signal is then used to mediate the strength of the LTP or LTD effect in the neuron.

Spike-Timing-Dependent-Plasticity (STDP) (Bi & Poo, 1998), incorporates temporal information by adjusting weights based on the timing of pre-synaptic and post-synaptic excitations

$$\Delta w = \begin{cases} A^+ \exp\left(\dfrac{t_{\text{pre}} - t_{\text{post}}}{\tau^+}\right), & \text{if } t_{\text{pre}} \leq t_{\text{post}}, \\ -A^- \exp\left(-\dfrac{t_{\text{pre}} - t_{\text{post}}}{\tau^-}\right), & \text{if } t_{\text{pre}} > t_{\text{post}}. \end{cases} \tag{4}$$

LTP / LDP is then achieved through the consistent temporally forward / backward spiking of a particular neuron. Due to the inherent ability of this plasticity rule to explode, regularization must be applied in order to constrain the growth scale of the weights.

$$A^+(w) = \eta^+ \exp(w_{\text{init}} - w), \tag{5}$$

$$A^-(w) = \eta^- \exp(w - w_{\text{init}}). \tag{6}$$

### 2.2 Hebbian Networks

Hebbian networks are generally simple to implement due to their lacking the backpropagation step. Soft-Hebb (Moraitis et al., 2022) uses an approach to Hebbian learning with a learning rule reminiscent of Oja's rule given by

$$\Delta W_{ij} = \alpha y_i (x_j - u_i W_{ij}) \tag{7}$$

where $u_j = (Wx)_i$ is the unactivated j-th hidden layer output. This network is trained as a winner-takes-all (WTA) network, which is realized through use of the softmax function with a temperature parameter $\tau$.

$$y_i = \frac{e^{\frac{u_i}{\tau}}}{\sum_{i'} e^{\frac{u_{i'}}{\tau}}} \tag{8}$$

The biological inspiration for this type of network comes from the concept of lateral inhibition (Douglas & Martin, 2004), which refers to the ability of neurons within a network to dampen the signals of lateral neurons. Lateral inhibition has been shown to occur in biological processing of visual, auditory, and tacticle information (Yantis & Abrams, 2017).

While the shallow SoftHebb networks perform well on simple tasks such as MNIST classification, it does not appear to be competitive with backpropagation on more complicated datasets such as CIFAR-10 and STL-10 (Journé et al., 2023), despite the addition of hidden layers into the network. Notably, these networks have been trained in an i.i.d setting, as opposed to the continual process that is more representative of biological learning.

## 2.3 Neuron-Wise Plasticity

As mentioned above, n-SNAP ties the plasticity of each input weight to a given neuron, such that $\Delta W_{ij}$ is only dependent on $i$. It's worthwhile to discuss biological motivations for this idea, and previous methods that implement this.

Synaptical scaling (Turrigiano et al., 1998) has been proposed as a method by which the strength of all synaptic inputs to a neuron are scaled as a function of the total activity of the neuron. This theory is a form of neuron specific homeostatis, which allows for a natural regulation of Hebb's rule (Hebb, 2005), which doesn't account for exploding weight values at each synapse. Tetzlaff et. al. (Tetzlaff, 2011) propose a mathematical formulation of this idea where the overall strength of the scaling factor is a function of the actual ($\nu$) and target ($\nu_T$) activity of a neuron.

$$\dot{w} \approx \mu G(w, u) + \gamma H(\nu^T - \nu) \tag{9}$$

Here, $G$ represents the plasticity factor, $H$ represents the scaling factor, $u$ is the neuron input, and $\mu$, $\gamma$ are constant factors.

von der Malsburg (1973) proposed a model for the visual cortices in which the total incoming synaptic strength to a neuron is constant. This model serves a twofold purpose, ensuring that a Hebbian system of excitatory synapses is stable, and ensuring that neurons become insensitive to stimuli that they are not learned on. In ANNs, this concept of constant total synaptic strength is realized through weight normalization (WN) (Salimans & Kingma, 2016), which reparameterizes weight vectors as a unit vector **v** and magnitude $g$. This parameterization has been shown to increase the speed of convergence when applied to SGD. Extensions of this method include centered WN (Huang et al., 2017b), projection based WN (Huang et al., 2017a), and constrained WN (Ikami et al., 2021).

## 2.4 Catastrophic Forgetting

The challenge of continual learning has been extensively studied, with various approaches proposed for networks with fixed capacity. These methods can generally be classified into three main categories: replay, regularization-based, and parameter isolation (Lange et al., 2021). Replay methods, inspired by the brain's episodic replay during sleep and rest, periodically revisit stored samples during or after learning a new task, effectively rehearsing previous knowledge (Kudithipudi et al., 2022). Previous attempts at avoiding catastrophic forgetting with Hebbian Learning have used pseudo-replay methods (Nakano & Hattori, 2017). Regularization-based methods draw inspiration from the brain's synaptic meta-plasticity, adjusting each

synaptic weight based on its estimated importance, as determined by other techniques (Kirkpatrick et al., 2017). Lastly, parameter isolation methods allocate specific model parameters to different tasks, allowing for specialization without interference (Serra et al., 2018).

| Method | Idea | Drawback |
|---|---|---|
| Replay | Periodically sample from prior learned tasks during training | Requires persistent storage of previous task examples. |
| Regularization | Penalize changes to important parameters during training | Estimation of important parameters can be suboptimal. |
| Parameter Isolation | Dedicate different parameters for each learned task | Cannot use shared output layer between tasks. |

Table 1: A short summary of prior methods for preventing catastrophic forgetting.

SNAP effectively uses the weight strength to determine importance and can be considered an implicit regularization-based method. To the best of our knowledge, it is the first method which does not require any additional machinery which explicitly keeps track of previous learning, like stored past data, pseudo-patterns, or past weights to prevent catastrophic forgetting. It does not require parameter estimation, and uses a shared output layer to classify each task. As such, it avoids the drawbacks seen in previous methods.

## 3 Methods

Let $W_{ij}$ be the synaptic weight linking presynaptic neuron $j$ to postsynaptic neuron $i$ in an ANN. Assume for the sake of simplicity that the weights are positive. Given that the weight is initialized with value $W_{ij}^{(0)}$ and given a learning algorithm and data which give a weight update $\delta W_{ij}^{(t)}$ at training step $t$, then we have that the weight at time $T$ $W_{ij}^{(T)}$ is given by

$$W_{ij}^{(T)} = W_{ij}^{(0)} + \sum_{t=1}^{T} \delta W_{ij}^{(t)}. \tag{10}$$

We define this as linear weight growth because the weight grows linearly in the $\delta W_{ij}^{(t)}$'s. This is represented in Fig. 1a.

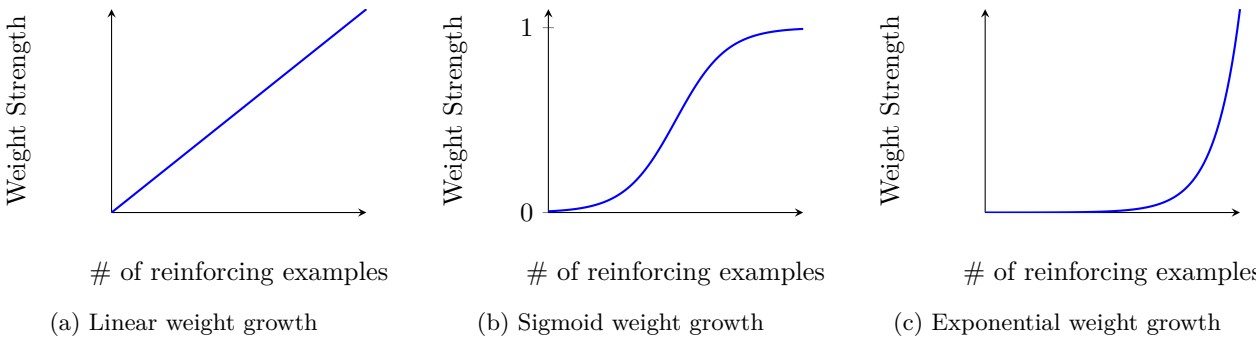

(a) Linear weight growth     (b) Sigmoid weight growth     (c) Exponential weight growth

Figure 1: Illustrations of different weight growth behaviours

In order to have LTP-inspired adaptive plasticity, we wish to modify the synaptic strength growth to have the following properties:

1. Limited Growth: Synaptic strength does not grow to infinity, but plateaus after reaching a large enough value.

2. Consolidation: Once the synaptic strength reaches a plateau, it is difficult for further training to affect its value.

There are many ways to achieve the above two properties, but having a weight growth behaviour which plateaus after reaching a certain value is the path we wish to explore in this paper. To implement this we choose the sigmoid function (Fig. 1b). This means that we wish to change equation 10 to

$$W_{ij}^{(T)} = \sigma \left( W_{ij}^{(0)} + \sum_{t=1}^{T} \delta W_{ij}^{(t)} \right),$$
(11)

where $\sigma(x) = \frac{1}{1+e^{-x}}$ is the sigmoid function, as represented diagrammatically in Fig. 1b. From equation 11 we have that

$$W_{ij}^{(T)} - W_{ij}^{(T-1)} \approx \left( 1 - W_{ij}^{(T-1)} \right) W_{ij}^{(T-1)} \delta W_{ij}^{(T)}.$$
(12)

Similarly, if instead of wanting sigmoidal growth we had wanted exponential weight growth, instead of equation 12 we would have gotten

$$W_{ij}^{(T)} - W_{ij}^{(T-1)} \approx W_{ij}^{(T-1)} \delta W_{ij}^{(T)}.$$
(13)

For more details on the derivation of equation 12 and equation 13 see A.6.

Accommodating negative weights by using absolute values in equation 12 we can turn any learning rule which provides weight changes $\delta W_{ij}$ into a rule which has sigmoidal weight growth by applying weight changes $\Delta W_{ij}$ instead where

$$\Delta W_{ij} = |W_{ij}|(1 - |W_{ij}|)\delta W_{ij}.$$
(14)

In the above, the plasticity of each synapse can be different and is given by $|W_{ij}|(1 - |W_{ij}|)$. Modifying the weight updates as in equation 14 gives us s-SNAP.

It may be useful to have the plasticity of all incoming synapses to a neuron be tied. Due to the fact that hidden layer neurons have been shown to act as feature selectors for input signals (Journé et al., 2023), it's reasonable to assume that each neuron could be primarily used to solve a specific task. To do this, we require a mapping from each neuron weight vector $W_i$ to a singular scalar value $f(W_i)$, which would represent the total synaptic strength of the neuron. As such, this scalar value should have the following properties

1. When the weight vector vanishes, the neuron cannot be excited anymore, so $f(W_i)$ should also vanish.

2. $f(W_i)$ should be nonnegative for any value of $W_i$, since it represents a total magnitude.

3. The value of $f(W_i)$ should always be bounded, as we cannot have infinite synapse strength.

A simple vector norm $f(W_i) = \|W_i\|_2$ fulfills all these requirements. We note that this is by no means a conclusive choice, there are multiple other forms that could be considered (equation 9 for example chooses $G$ to be a quadratic function of $W_i$). Our choice was made for simplicity and a reduction in the number of degrees of freedom of our system.

With this mapping, the weight change update in equation 14 would be changed to

$$\Delta W_{ij} = \|W_i\|_2 (1 - \|W_i\|_2)\delta W_{ij}.$$
(15)

In the above, the plasticity of each input synapse at neuron $i$ is the same and is given by $\|W_i\|_2 (1 - \|W_i\|_2)$. Modifying the weight updates as in equation 15 gives us n-SNAP.

|              | **Linear**                     | **Sigmoidal**                                    | **Exponential**                        |
| ------------ | ------------------------------ | ------------------------------------------------ | -------------------------------------- |
| **Synapse-wise** | $\Delta W_{ij} = \delta W_{ij}$ | $\Delta W_{ij} = \lvert W_{ij}\rvert(1 - \lvert W_{ij}\rvert)\delta W_{ij}$ | $\Delta W_{ij} = \lvert W_{ij}\rvert\delta W_{ij}$ |
| **Neuron-wise**  | $\Delta W_{ij} = \delta W_{ij}$ | $\Delta W_{ij} = \lVert W_i\rVert_2(1 - \lVert W_i\rVert_2)\delta W_{ij}$ | $\Delta W_{ij} = \lVert W_i\rVert_2\delta W_{ij}$ |

Table 2: Given a learning rule which given weight updates $\delta W_{ij}$, applying the modified weight updates $\Delta W_{ij}$ instead, will allow for linear, sigmoidal, or exponential growth in either the weight values (synapse-wise) or in the input-weight norms (neuron-wise).

## 4 Experiments

### 4.1 Experiment 1: I.I.D. Data

We train an MLP with one hidden layer (cf. section A.2 for more details on the architecture) using either Hebbian Learning (cf. section A.5 for more details on our implementation of Hebbian Learning) or SGD with cross-entropy loss, on i.i.d. datasets (MNIST and FashionMNIST). We vary the type of weight growth by layer, and denote these layers using LIN, SIG, and EXP respectively. For example, a network with a sigmoid growth hidden layer and linear growth classification layer will be denoted by (SIG/LIN).

For Hebbian Learning, we can see from Fig. 2 that as long as we choose the right hyperparameter $\lambda$ (which controls the strength of lateral inhibition cf. section A.5 for more details) all types of weight growth can performe roughly equally well in the i.i.d. setting. The same is true for the SGD trained network (cf. table 7).

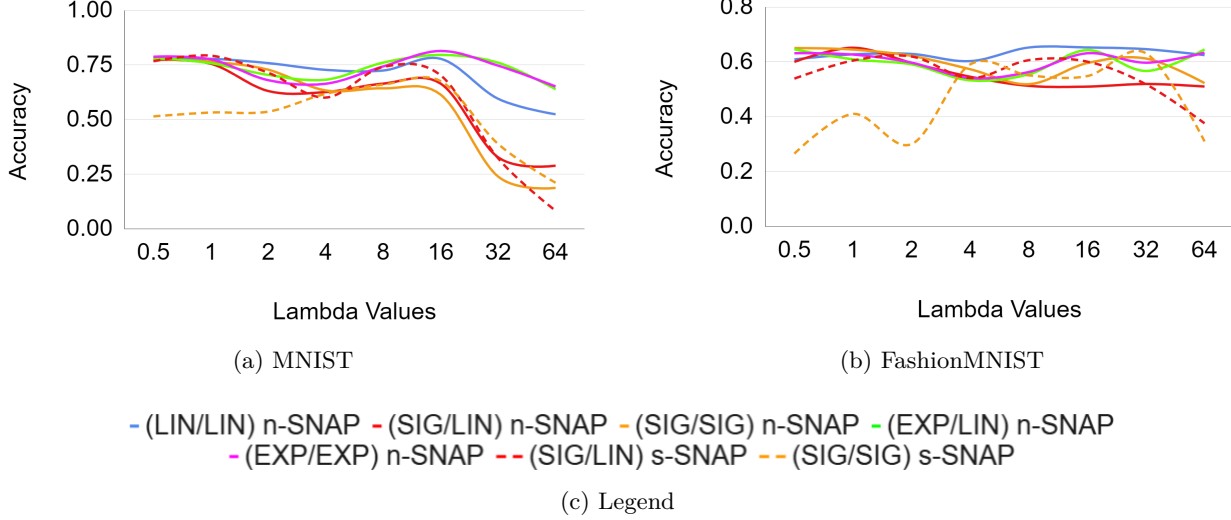

(a) MNIST  (b) FashionMNIST

- (LIN/LIN) n-SNAP  - (SIG/LIN) n-SNAP  - (SIG/SIG) n-SNAP  - (EXP/LIN) n-SNAP
- (EXP/EXP) n-SNAP  -- (SIG/LIN) s-SNAP  -- (SIG/SIG) s-SNAP

(c) Legend

Figure 2: Accuracy of Hebbian MLP on **i.i.d. datasets** MNIST and FashionMNIST as a function of lateral inhibition hyperparameter $\lambda$. The solid lines denote n-SNAP while the dotted lines denote s-SNAP (cf. table 2).

### 4.2 Experiment 2: Sequential Task Learning

To test sequential task learning we turn both MNIST and FashionMNIST into sequential tasks by sequentially training the model on five tasks. Task 1 trains only on images of classes 0 and 1, task 2 trains only on images of classes 2 and 3, and so forth. The model switches to the next task when it reaches a threshold accuracy $a_{thr}$, or 35 epochs, whichever comes first. The specific value of $a_{thr}$ is chosen such that the network reaches close to maximal experimental accuracy on a task before beginning a new one. Fig. 3 shows the average test accuracy across all classes (0-9) after training on all tasks sequentially for Hebbian models with different lateral inhibition strengths.

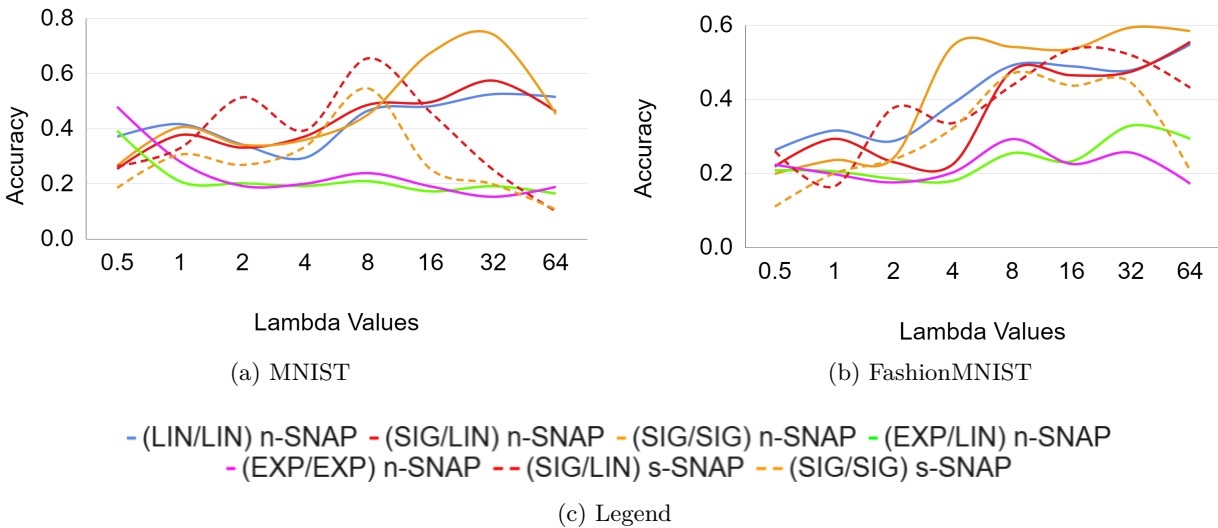

(a) MNIST

(b) FashionMNIST

(c) Legend

Figure 3: Average accuracy of Hebbian MLP accross all classes after learning via the **sequential task** versions of MNIST and FashionMNIST as a function of lateral inhibition hyperparameter $\lambda$ and the type of weight growth.

Models with sigmoidal weight growth in their hidden layers achieve the highest average test accuracies by reducing catastrophic forgetting through a consolidation phase, as shown in Figure 3. Exponential growth models, despite excelling in I.I.D. experiments, perform the worst due to rapid learning and forgetting, while linear growth models show average performance with moderate forgetting.

Figure 4 presents the best-performing Hebbian models on sequential MNIST learning (cf. Fig. 7 for the same results on FashionMNIST) for various growth types. Catastrophic forgetting can be seen to occur using SNAP when using exponential and linear weight growth, 4d and 4b. Importantly, we can see from Fig. 4d and 7d that even the best linear growth Hebbian learning networks also suffer from catastrophic forgetting, even if it is not as severe as for SGD trained models. Figures 4a and 4c show (SIG/LIN) and (SIG/SIG) n-SNAP networks that are able to successfully prevent catastrophic forgetting in the Hebbian network. Figure 7b shows a similar though weaker result on the FashionMNIST dataset.

Figure 5 applies the best performing network architecture, (SIG/SIG), to an SGD model on the sequential MNIST task. While SNAP was effective in preventing catastrophic forgetting in Hebbian networks, the same cannot be said of SGD trained networks. A clear inability to sustain performance on prior tasks is seen, even when (SIG/SIG) n-SNAP, which was the most effective in prevention on the Hebbian networks, is applied. Looking into why SNAP struggles when applied to SGD, it appears to be due to the network reaches high accuracies before the weights have grown to maximal size. This premature learning of the digits does not allow for the weights to enter into the flat consolidation phase of the sigmoid curve, such that when a new task is started, weights that have been learned for a prior task will still be highly plastic, causing the forgetting that is seen throughout all of Figure 5.

## 5   Discussion

In this paper, we introduced SNAP, a biologically inspired mechanism in order to prevent catastrophic forgetting in sequential task learning. We investigated various types of weight growth at different layers of the network and introduced two forms of SNAP, s-SNAP and n-SNAP, which applies SNAP at the synaptic and neuronal levels respectively. Sigmoidal weight growth, inspired by the biological concept of long term potentiation, was shown to be the most effective at perserving performance on prior tasks when switching to novel ones.

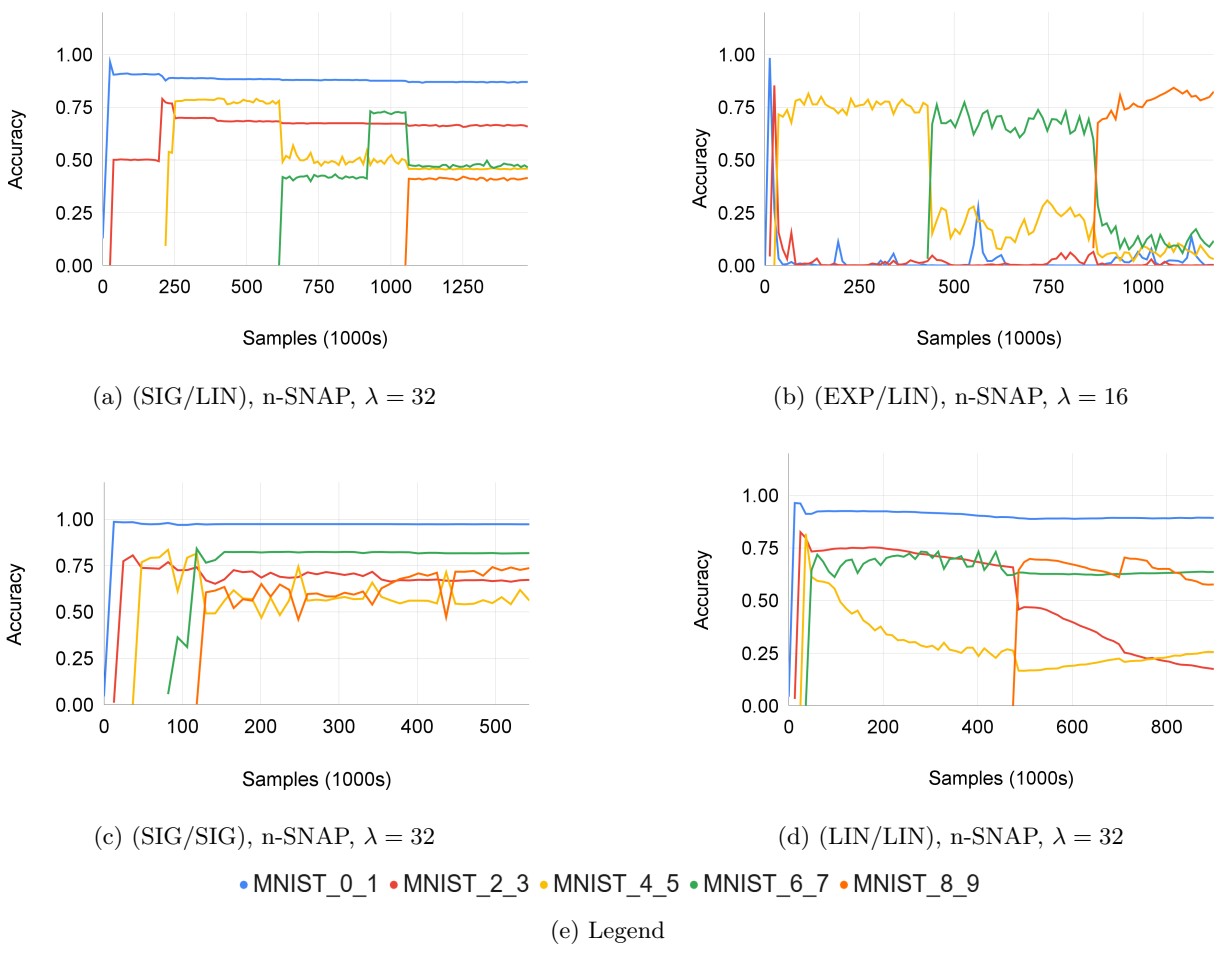

(a) (SIG/LIN), n-SNAP, $\lambda = 32$

(b) (EXP/LIN), n-SNAP, $\lambda = 16$

(c) (SIG/SIG), n-SNAP, $\lambda = 32$

(d) (LIN/LIN), n-SNAP, $\lambda = 32$

• MNIST_0_1 • MNIST_2_3 • MNIST_4_5 • MNIST_6_7 • MNIST_8_9

(e) Legend

Figure 4: Comparison of top-performing Hebbian models on sequential MNIST task learning for different hidden and classification layer weight growth types. Each line represents retention of test accuracy across tasks.

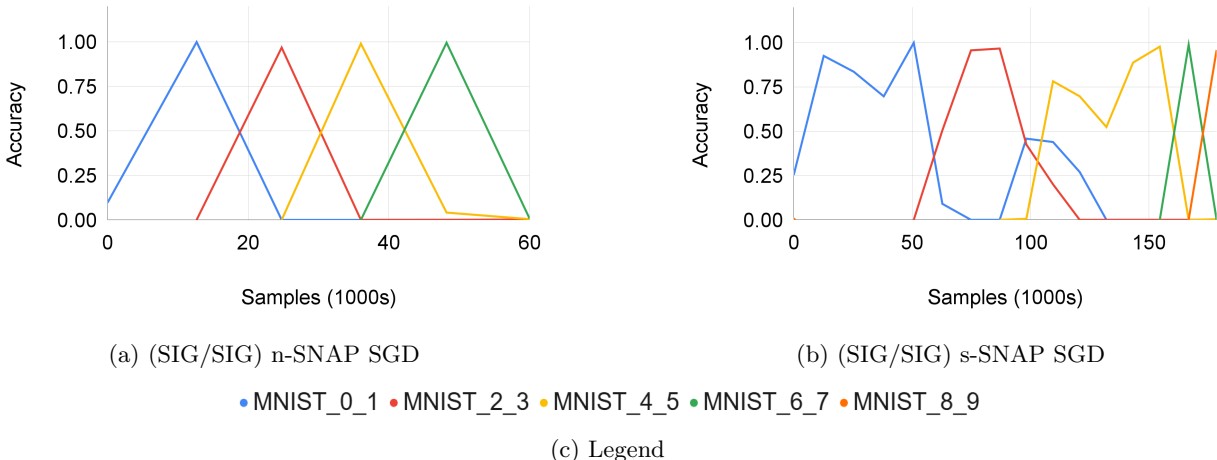

(a) (SIG/SIG) n-SNAP SGD

(b) (SIG/SIG) s-SNAP SGD

• MNIST_0_1 • MNIST_2_3 • MNIST_4_5 • MNIST_6_7 • MNIST_8_9

(c) Legend

Figure 5: Comparison of top-performing sigmoidal SGD models on sequential MNIST task learning.

Hebbian learning naturally suffers slightly less than SGD-trained networks from catastrophic forgetting, but still suffers from it enough that it cannot train successfully on many sequential tasks. However, when trained with sigmoidal n-SNAP, we have signifcant protection against catastrophic forgetting of prior tasks. The same cannot be said of SGD, while helping slightly with catastrophic forgetting for sigmoidal weight, does not prevent it in this case, with the top-performing models with both n-SNAP and s-SNAP showing a significant degradation in performance across tasks.

## 6 Limitations

This paper uses a simple mathematical formulation of LTP in artifical networks by employing sigmoidal weight growth in order to constrain and consolidate the synaptical strength in the network. It remains a question as to what functional form would be best to represent LTP, with one notable weakness of the sigmoid function being the slow initial growth, requiring a large number of training samples to reach the flat consolidation portion of the curve. A weight growth curve with an initial exponential-like growth would allow for consolidation with less samples shown, and faster learning of the model across multiple tasks.

The experiments conducted in this paper was limited to the baseline MNIST and FashionMNIST datasets. This scope should be widened to include more complicated datasets such as CIFAR-100 or ImageNet in order to increase confidence in these results. Further research can also extend this sort of weight growth to more biologically inspired learning algorithms such as the Wake-Sleep (Hinton et al., 1995) or Forward-Forward (Hinton, 2022) algorithms to see if the results generalize.

## 7 Acknowledgments

This research was supported by a grant from Strong Compute who generously provided us with compute.

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

# A Appendix

## A.1 Code

All code can be found at `https://github.com/lshiyan/biological-deep-learning`.

## A.2 Neural Net Architecture

Our models follow a simple architecture, (Figure 6), that consists of three layers: input, hidden, and output.

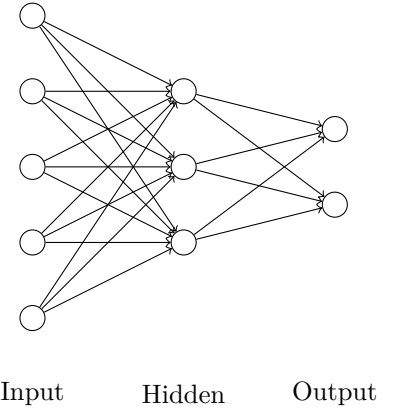

Input      Hidden      Output

Figure 6: Simplified Model Configuration

## A.3 Notation Introduction

| Category | Symbol | Description |
|---|---|---|
| Hidden Layer | $x_i$ | Activity of the presynaptic neuron. |
| | $a_i$ | Activity of the postsynaptic neuron. |
| | $r_{ij}$ | Computed result term from the learning rule, used for weight updates. |
| | $h_i$ | Post lateral inhibition activation of each neuron in the hidden layer. |
| Classification Layer | $\hat{y}_i$ | Ground truth label for the sample. |
| | $\tilde{y}_i$ | Predicted label by the model. |
| | $h_i$ | Post lateral inhibition activation from the hidden layer, serving as input. |
| Weight Update | $\delta W_{ij}$ | Change in weights. |
| | $\alpha$ | Learning rate. |
| | $f(x)$ | Function used for weight growth (e.g., linear, sigmoid, exponential). |

Table 3: Notation for Hidden Layer, Classification Layer, and Weight Update

## A.4 Hyperparameter Specification

### I.I.D. Experiments Hyperparameters

For all experiments, our models use the hyperparameters specified in Table 4. We evaluated the optimal values of $\eta$ (as defined in Equation 20) and $\alpha$ for seven weight growth model configurations: linear-linear, sigmoid-linear neuron, sigmoid-linear synapse, exponential-linear neuron, exponential-exponential neuron,

sigmoid-sigmoid neuron, and sigmoid-sigmoid synapse. These evaluations were conducted on both MNIST and FashionMNIST datasets.

In these configurations, the first term refers to the growth function applied to the hidden layer, and the second term to the output layer. Growth can be either neuron-wise (applied across all synapses of a neuron) or synapse-wise (applied individually to each synapse). For example, "sigmoid-linear neuron" means sigmoid growth for the hidden layer and linear growth for the output layer, applied on a neuron-wise basis.

For each configuration, we performed the I.I.D. experiments over 10 epochs to identify the best $\eta$ and $\alpha$ pairs based on test accuracy. The identified optimal pairs were then used consistently across all experiments, including both the I.I.D. and Sequential Task experiments, without further re-tuning for the Sequential Task experiments.

The values of $\eta$ tested were 0.7, 0.3, 0.1, 0.03, and 0.001. The values of $\alpha$ tested were 0.3, 0.1, 0.03, 0.01, and 0.003.

The optimal $\eta$ and $\alpha$ pairs for MNIST are detailed in Table 6, while the corresponding values for FashionMNIST are presented in Table 5.

| Hyperparameter | MNIST Experiment | FashionMNIST Experiment |
|---|---|---|
| Input Dimension | 784 | 784 |
| Hidden Layer Dimension | 64 | 96 |
| Output Layer Dimension | 10 | 10 |
| Beta | 0.01 | 0.01 |
| Initialization | Uniform(0,beta) | Uniform(0,beta) |
| Batch Size | 1 | 1 |
| Epochs | 10 | 10 |

Table 4: General Model Hyperparameters for MNIST and FashionMNIST Experiments

| λ / Model | L.L. | E.L. Neuron | S.L. Neuron | S.S. Neuron | E.E. Neuron | S.L. Synapse | S.S. Synapse |
|---|---|---|---|---|---|---|---|
| 0.5 | (0.03, 0.01) | (1, 0.001) | (1, 0.01) | (0.3, 0.1) | (1, 0.001) | (1, 0.001) | (0.1, 0.001) |
| 1 | (0.1, 0.01) | (1, 0.01) | (0.3, 0.01) | (0.3, 0.03) | (1, 0.001) | (1, 0.001) | (1, 0.001) |
| 2 | (0.1, 0.001) | (1, 0.001) | (0.1, 0.01) | (0.3, 0.001) | (1, 0.03) | (1, 0.01) | (0.3, 0.03) |
| 4 | (0.03, 0.001) | (1, 0.001) | (0.3, 0.001) | (0.3, 0.001) | (1, 0.001) | (0.7, 0.001) | (0.3, 0.001) |
| 8 | (0.01, 0.01) | (1, 0.001) | (0.3, 0.001) | (0.3, 0.001) | (1, 0.001) | (0.1, 0.03) | (0.3, 0.03) |
| 16 | (0.01, 0.03) | (1, 0.01) | (0.3, 0.001) | (0.7, 0.001) | (1, 0.01) | (0.3, 0.1) | (0.7, 0.01) |
| 32 | (0.01, 0.01) | (0.3, 0.1) | (0.7, 0.1) | (0.7, 0.3) | (0.3, 0.03) | (0.7, 0.1) | (0.7, 0.1) |
| 64 | (0.01, 0.01) | (0.1, 0.1) | (0.7, 0.3) | (0.01, 0.1) | (0.1, 0.03) | (0.3, 0.3) | (0.01, 0.001) |

Table 5: Optimal $\eta$ and learning rate pairs for various weight growth model configurations on the FashionM-NIST dataset. These values represent the best-performing parameters from the I.I.D. classification experiments and were used consistently across all experiments, including the sequential task learning experiment. Here, L represents linear, S represents sigmoid, and E represents exponential.

## A.5 Hebbian Learning

### A.5.1 Lateral Inhibition in Feedforward Propagation

During forward propagation, lateral inhibition—a mechanism where excited neurons suppress their neighbors—promotes sparse, distinct activations, enhancing contrast in stimuli. Our model uses equation equation 17, where increasing the $\lambda$ parameter results in sparser hidden layer neuron activations.

$$a_i = \text{ReLU}\left(\sum_j W_{ij} x_j\right) \tag{16}$$

| λ / Model | L.L. | E.L. Neuron | S.L. Neuron | S.S. Neuron | E.E. Neuron | S.L. Synapse | S.S. Synapse |
|-----------|------|-------------|-------------|-------------|-------------|--------------|--------------|
| **0.5** | (0.03, 0.01) | (1, 0.001) | (1, 0.01) | (0.3, 0.1) | (1, 0.001) | (1, 0.001) | (0.1, 0.001) |
| **1** | (0.1, 0.01) | (1, 0.01) | (0.3, 0.01) | (0.3, 0.03) | (1, 0.001) | (1, 0.001) | (1, 0.001) |
| **2** | (0.1, 0.001) | (1, 0.001) | (0.1, 0.01) | (0.3, 0.001) | (1, 0.03) | (1, 0.01) | (0.3, 0.03) |
| **4** | (0.03, 0.001) | (1, 0.001) | (0.1, 0.01) | (0.3, 0.001) | (1, 0.001) | (1, 0.03) | (0.1, 0.1) |
| **8** | (0.01, 0.01) | (1, 0.001) | (0.1, 0.03) | (0.1, 0.1) | (1, 0.001) | (0.1, 0.001) | (0.1, 0.001) |
| **16** | (0.01, 0.03) | (1, 0.01) | (0.003, 0.01) | (0.003, 0.3) | (1, 0.01) | (1, 0.01) | (1, 0.1) |
| **32** | (0.01, 0.01) | (0.3, 0.1) | (0.01, 0.01) | (0.01, 0.3) | (0.3, 0.03) | (0.1, 0.3) | (1, 0.1) |
| **64** | (0.01, 0.01) | (0.1, 0.1) | (0.001, 0.1) | (0.003, 0.3) | (0.1, 0.03) | (0.03, 0.03) | (0.3, 0.3) |

Table 6: Optimal $\eta$ and learning rate pairs for various weight growth model configurations on the MNIST dataset. These values represent the best-performing parameters from the I.I.D. classification experiments and were used consistently across all experiments, including the sequential task learning experiment. Here, L represents linear, S represents sigmoid, and E represents exponential.

$$h_i = \left( \frac{a_i}{\max_k(a_k)} \right)^\lambda \tag{17}$$

### A.5.2 Weight Update Mechanism

Most existing approaches train neural network layers sequentially: first, the initial layer is trained on all the data, and then the outputs from this trained layer are used as inputs to the next layer, and so on. While this method may simplify training, we do not consider it realistic or practical for real-world applications where simultaneous learning is often required. Therefore, in our approach, we train all layers simultaneously, allowing the network to learn in a more integrated and biologically relevant manner.

In what follows, (pre-SNAP) weight updates use equation 18 to update their weights.

$$\delta W_{ij} = \alpha r_{ij}, \tag{18}$$

where $\alpha$ is the learning rate and the learning rule prescribes $r_{ij}$.

**Hidden Layer Weight Update Rule:** The hidden layer employs Sanger's rule equation 20(Sanger, 1989), which extends the basic Hebbian Learning rule introduced in Equation equation 19(Krotov & Hopfield, 2016). Basic Hebbian learning updates weights based on the product of the presynaptic activity $x_j$ and postsynaptic activity $h_i$, reinforcing the connections between co-active neurons.

$$r_{ij} = h_i \cdot x_j \tag{19}$$

Sanger's rule builds on this by aiming to make the neurons represent orthogonal features. It sequentially extracts principal components by subtracting the projection onto previously extracted components, as shown in Equation equation 20. The term $\sum_{k=1}^{i-1} h_k w_{kj}$ represents the projection onto the previous outputs, ensuring each neuron's weight vector remains orthogonal to those of prior neurons.

$$r_{ij} = h_i x_j - \eta h_i \sum_{k=1}^{i-1} h_k w_{kj} \tag{20}$$

where $\eta$ controls the strength of the orthogonality constraint in Sanger's rule.

**Classification Layer Weight Update Rule:** The classification layer in our model employs a supervised Hebbian Learning rule, as defined in Equation equation 21. This approach contrasts with the traditional

use of SGD, which is commonly used in other works. Our supervised Hebbian rule offers a novel and more biologically plausible alternative to the standard SGD-based methods, enhancing the neural network's alignment with biological learning principles.

$$r_{ij} = (\hat{y}_i - \tilde{y}_i)x_j, \tag{21}$$

### A.6 Derivation of Weight Growth Updates

Assuming that we have

$$W_{ij}^{(T)} = \sigma\left(W_{ij}^{(0)} + \sum_{t=1}^{T}\delta W_{ij}^{(t)}\right), \tag{22}$$

where $\sigma(x) = \frac{1}{1+e^{-x}}$ is the sigmoid function. Then we have that

$$W_{ij}^{(T)} - W_{ij}^{(T-1)} \quad \approx \sigma'\left(W_{ij}^{(0)} + \sum_{t=1}^{T-1}\delta W_{ij}^{(t)}\right)\delta W_{ij}^{(T)} \tag{23}$$

$$= \left(1 - \sigma\left(W_{ij}^{(0)} + \sum_{t=1}^{T-1}\delta W_{ij}^{(t)}\right)\right)\sigma\left(W_{ij}^{(0)} + \sum_{t=1}^{T-1}\delta W_{ij}^{(t)}\right)\delta W_{ij}^{(T)} \tag{24}$$

$$= \left(1 - W_{ij}^{(T-1)}\right)W_{ij}^{(T-1)}\delta W_{ij}^{(T)}, \tag{25}$$

where equation 23 is the first order expansion in the Taylor series, equation 24 uses the identity that the derivative of the sigmoid is $\sigma'(x) = (1 - \sigma(x))\sigma(x)$, and equation 25 comes from using equation 22.

Similarly, if instead we wish to have exponential weight growth such that

$$W_{ij}^{(T)} = \exp\left(W_{ij}^{(0)} + \sum_{t=1}^{T}\delta W_{ij}^{(t)}\right). \tag{26}$$

Then we have that

$$W_{ij}^{(T)} - W_{ij}^{(T-1)} \quad \approx \exp'\left(W_{ij}^{(0)} + \sum_{t=1}^{T-1}\delta W_{ij}^{(t)}\right)\delta W_{ij}^{(T)} \tag{27}$$

$$= \exp\left(W_{ij}^{(0)} + \sum_{t=1}^{T-1}\delta W_{ij}^{(t)}\right)\delta W_{ij}^{(T)} \tag{28}$$

$$= W_{ij}^{(T-1)}\delta W_{ij}^{(T)}, \tag{29}$$

where equation 27 is the first order expansion in the Taylor series, equation 28 uses the identity that the derivative of the exponential is the exponential, and equation 29 comes from using equation 26.

### A.7 SGD: Effect of Weight Growth on Forgetting

In order to see if having a sigmoidal weight growth could also help mitigate catastrophic forgetting when training with SGD, we repeated all our experiments but this time the $\delta W_{ij}$ in Table 2 were obtained using SGD on cross-entropy loss rather through Hebbian learning. The network architecture stayed the same and the applied weight changes were still the $\Delta W_{ij}$ of Table 2.

As can be seen from Fig. 5, having sigmoidal weight growth does not prevent catastrophic forgetting when learning with SGD, though it slows down the forgetting slightly compared to the usual linear weight growth.

| | (EXP/EXP) n-SNAP | (SIG/SIG) n-SNAP | (LIN/LIN) n-SNAP |
|---|---|---|---|
| **MNIST** | 0.9147 | 0.9491 | 0.9731 |
| **FashionMNIST** | 0.8333 | 0.8661 | 0.8717 |

Table 7: Classification accuracies of SGD for different weight growths for MNIST and FashionMNIST dataset in the i.i.d. setting.

## A.8  Supplementary Graphs

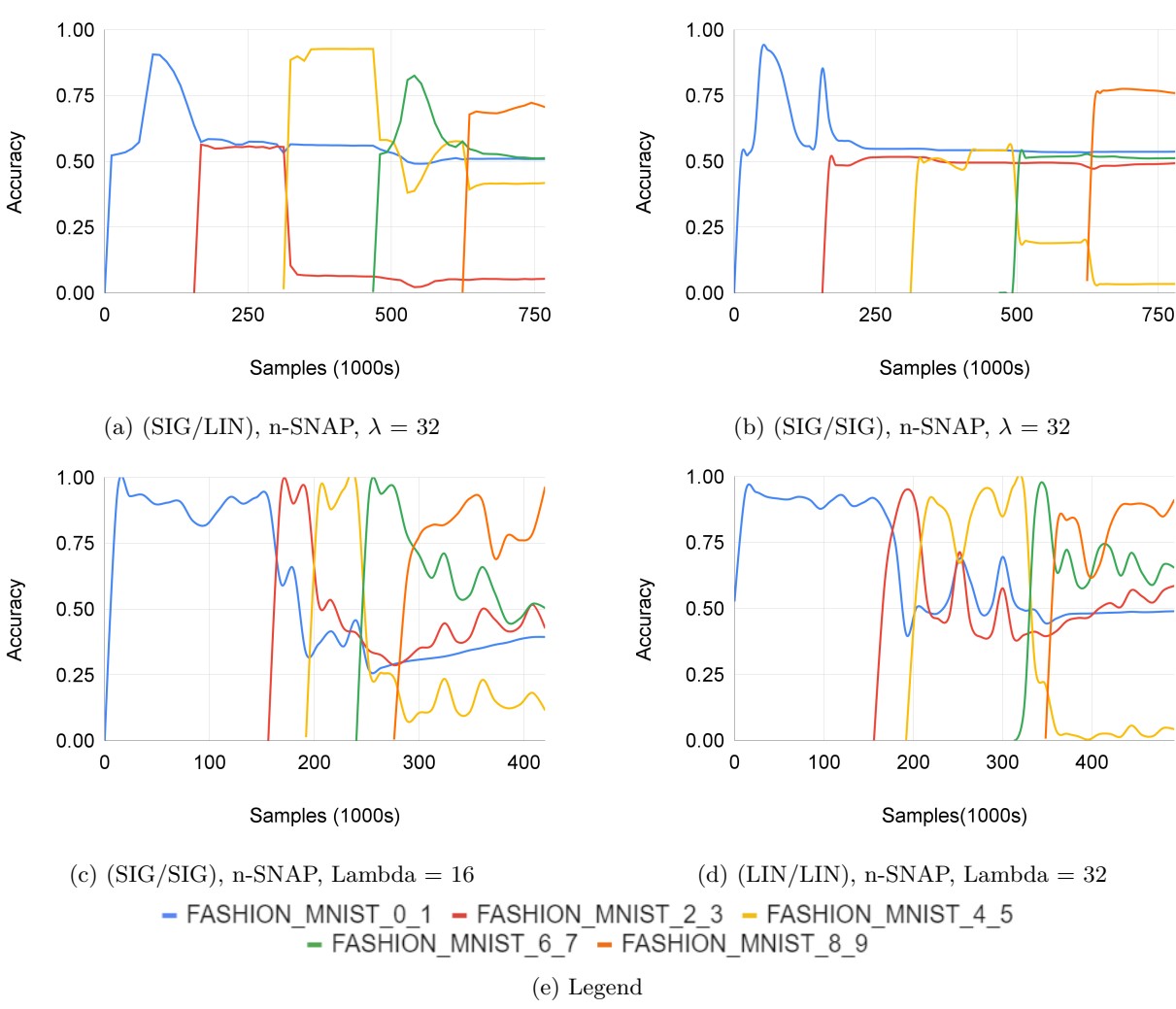

(a) (SIG/LIN), n-SNAP, $\lambda = 32$

(b) (SIG/SIG), n-SNAP, $\lambda = 32$

(c) (SIG/SIG), n-SNAP, Lambda = 16

(d) (LIN/LIN), n-SNAP, Lambda = 32

FASHION_MNIST_0_1    FASHION_MNIST_2_3    FASHION_MNIST_4_5
FASHION_MNIST_6_7    FASHION_MNIST_8_9

(e) Legend

Figure 7: Comparison of top-performing **Hebbian** models on **sequential task learning experiments** for different hidden and classification layer weight growth functions in **FashionMNIST**. **Legend:** Each line represents the retention of test accuracy across different tasks. **Tasks:** The blue line indicates the test accuracy on the first task (e.g., **FashionMNIST** classes 0 and 1), while the orange line shows the test accuracy on the last task