# OpenReview forum: "SNAP: Stopping Catastrophic Forgetting in Hebbian Learning with Sigmoidal Neuronal Adaptive Plasticity"
_TMLR — Rejected by TMLR_

### Review · Reviewer_Qz3z · 2026-01-14

**Summary Of Contributions:**

Catastrophic forgetting refers to the phenomenon where artificial neural networks forgets previously learned new tasks while they trained on new data. To address this problem, this paper adopts Hebbian learning, a biologically plausible learning paradigm, for continual learning instead of the standard backpropagation-based approach. To achieve this, the authors introduce SNAP (Sigmoidal Neuronal Adaptive Plasticity), a mechanism that modifies weight updates such that synaptic strengths follow a sigmoidal growth curve, mimicking Long-Term Potentiation (LTP) observed in biological neurons.. The key idea is that the initial synaptic weights update rapidly and gradually consolidate,  become resistant to change as they approach saturation. In contrast, both conventional Hebbian learning and SGD-based methods rely on linear weight updates, where plasticity remains constant regardless of synaptic strength. The author introduces two variants of SNAP: synapse-wise SNAP (s-SNAP), where plasticity depends on individual weights, and neuron-wise SNAP (n-SNAP), where plasticity depends on the norm of a neuron’s incoming weights. Finally, the authors evaluate two variants of SNAP on MNSIT and FashionMNIST datasets using hebbian and SGD methods and, they observe that sigmoidal weight growth effectively prevents catastrophic forgetting for Hebbian learning, without replay or explicit regularization, but does not fully solve forgetting for SGD-trained models.

**Contributions:**

* **Proposed SNAP (Sigmoidal Neuronal Adaptive Plasticity):** This study introduces a novel weight growth algorithm inspired by Long Term Potentiation in biological brains, where synaptic plasticity decreases as weights saturate.
* **Two variants of SNAP:** The authors propose and formalize both synapse-wise (s-SNAP) and neuron-wise (nSNAP) adaptive plasticity.
* **Comprehensive evaluation:** The study conducts an extensive evaluation of both SNAP variants on MNIST and FashionMNIST under i.i.d. and sequential (continual) learning settings, using both Hebbian learning and SGD-based optimization.

**Strengths:**

I found this work to have the following strengths:

**Clarity:** The introduction clearly defines catastrophic forgetting and Hebbian learning, and provides an explanation of how synaptic weight strengths evolve and eventually reach a consolidation threshold. The paper presents the theoretical motivation well, including detailed discussions of neuron-wise plasticity, Hebbian networks, catastrophic forgetting, and long-term potentiation, supported by clear mathematical formulations. The experimental setup and validation on two standard datasets are also described in a structured manner.

However, despite the clarity of the methodological presentation, the paper lacks limited evaluation, insufficient results discussion, limited related works which is clearly missing in current study.

**Originality:** The idea of using SNAP to model biologically plausible synaptic consolidation within Hebbian learning is conceptually novel. Furthermore, the distinction between synapse-wise SNAP (s-SNAP), where plasticity is defined at the level of individual weights, and neuron-wise SNAP (n-SNAP), where plasticity depends on the collective strength of a neuron’s incoming weights, is a meaningful and novel design choice. Evaluating and contrasting both variants within the same framework provides additional novelty and helps clarify the role of plasticity granularity in Hebbian continual learning.

**Significance:** This work is significant in that it contributes to catastrophic forgetting which is a challenging problem in continual learning when trying to learn new data. By introducing sigmoidal neuronal adaptive plasticity (SNAP) and evaluating two plasticity variants on standard benchmarks, the authors demonstrate that Hebbian learning combined with neuron-wise sigmoidal plasticity (n-SNAP) can provide substantial protection against catastrophic forgetting of previously learned tasks. Although the empirical evaluation is limited in scope, the results suggest that biologically inspired plasticity mechanisms can play an important role in continual learning, highlighting a promising direction for future research

**Audience:**

Yes

**Audience Explanation:**

Yes. The findings of this paper would be of interest to a meaningful portion of TMLR’s journal.

**Claims And Evidence:**

No

**Claims Explanation:**

From my perspective, the current submission provides preliminary evidence in support of its claims, but the results are not yet sufficiently convincing or clearly analyzed. Below, I outline the main weaknesses of the paper.

* **Limited evaluation and discussion on results:** Although the paper introduced two variants of SNAP, the majority of the paper majorly focused on related work and methods, while the experimental evaluation remains limited. The results section considers both i.i.d. and sequential learning settings; however, the main findings, implications, and comparative insights are not clearly presented. As a result, it is difficult to identify the key takeaways of the paper. Although Figs 3, 4 and 5 provide some interesting behaviors of SNAP under Hebbian and SGD-based learning, these results are largely left unexplained and insufficiently discussed. In particular, the paper does not clearly analyze why SNAP catastrophic forgetting is avoided in Hebbian learning but fails to prevent catastrophic forgetting under SGD, nor does it provide an explicit comparison between the two SNAP variants across settings. I strongly recommend that the authors substantially expand the discussion of experimental results, explicitly summarizing the key findings and their implications for each learning setting (i.i.d. vs. sequential, Hebbian vs. SGD), and clarifying how these results support the paper’s central claims.

* **Lack of discussion on hyperparameters and their impact on performance:** The SNAP method uses lateral inhibition hyperparameter (λ), however authors have not discussed the impact of (λ) on catastrophic forgetting in model training although authors presented in  Fig3 and Fig 4 plots. Additionally, the figures vary the number of training samples and task transitions, yet no analysis is provided to explain the observed trends. Without any discussion of these results, it is difficult to interpret the robustness and generality of the proposed method. I strongly encourage the authors to explicitly analyze the impact of key hyperparameters, especially λ, and to discuss the observations, conclusions, and implications suggested by the figures.

* **Limited related work and missing citations:** The authors main objective in this paper is to mitigate catastrophic forgetting in continuous learning using hebbian learning. However, the coverage of related work in this direction is limited and incomplete. Several important perspectives and research threads relevant to the paper’s motivation and contributions are missing.
    * First, the paper does not present how catastrophic forgetting display across different domains of continual learning (e.g., vision, control, representation learning), nor does it contextualize the severity of the problem in these settings.
    * Second, prior studies that use Hebbian or Hebbian-like learning rules to address catastrophic forgetting are only briefly mentioned, leaving an incomplete picture of existing approaches and their limitations.
    * Third, the manuscript lacks a discussion of known failure modes of biologically plausible learning algorithms when applied to ANNs, which is essential for understanding the trade-offs of the proposed method.
    * Fourth, while the paper is motivated almost exclusively by Hebbian learning, it does not justify this choice relative to other biologically plausible learning paradigms, such as Echo State Networks, equilibrium-based learning, or energy-based and local learning rules. A clearer explanation of why Hebbian learning is the most appropriate or relevant framework for SNAP would strengthen the paper’s positioning.
   * Overall, the introduction and related work sections would benefit from a broader and more structured discussion, explicitly situating SNAP within the motivation of biologically plausible continual learning methods, and by including appropriate citations to prior work in these areas.

Several relevant citations appear to be missing.

Kemker et al. 2018, Measuring catastrophic forgetting in neural networks, AAAI-2018

Lesort et al. 2020, Continual learning for robotics: Definition, framework, learning strategies, opportunities and challenges, Information fusion 2020

Merlin et al. 2023, WHAT HAPPENS DURING FINETUNING OF VISION TRANSFORMERS: AN INVARIANCE BASED INVESTIGATION, CoLLAs 2023

Camilla et al. 2025, Large Language Models as Model Organisms for Human Associative Learning, NeurIPS-2025

* **Lack of motivation and limited evaluation of datasets:** Although the authors main objective is clear in this paper, the paper does not sufficiently motivate why Hebbian learning is the appropriate framework for addressing catastrophic forgetting in ANNs.
     * First,  prior work has shown that catastrophic forgetting can arise in multiple settings, including during weight fine-tuning, class-incremental learning, domain shifts, and the addition of new tasks. In contrast, the current work primarily focuses on weight-update level forgetting, without clearly positioning this choice relative to other forms of catastrophic forgetting.
    * Second, the experimental evaluation is limited to MNIST and FashionMNIST, which are small-scale, image-only benchmarks. While suitable for initial validation, these datasets are insufficient to demonstrate the generality of the proposed method. Extending the evaluation to more challenging vision datasets (e.g., CIFAR-10/100, ImageNet-100) and ideally to other modalities such as text or speech would significantly strengthen the empirical claims.

Addressing these issues would substantially improve both the motivation and the empirical rigor of the paper.

For a complete and detailed account of both major issues, please refer to the “Questions” section.

**Requested Changes:**

Specifically, there are several points that I believe require further attention/work. I have divided these into major issues, which should be prioritized, and minor ones, which should be addressed for a strong version of current work.

**Major Comments/Questions**
* **Limited architecture choice:** The proposed SNAP method is currently evaluated on simple shallow MLP architecture with a single hidden layer. While this choice is reasonable for initial validation, it significantly limits the conclusions that can be drawn about the generality and scalability of the method. Modern continual learning challenges typically involve deeper architectures such as CNNs, RNNs and Transformer models, where catastrophic forgetting occurs differently due to hierarchical representations and parameter sharing across layers. This pose several questions:
  * Did the authors evaluate SNAP on deeper architectures beyond a shallow MLP with a single hidden layer?
  * Is there any discussion or justification for the choice of this architecture, and how SNAP could be extended to deeper models?
* **Which plasticity mechanism is most important?** While the paper empirically demonstrates that neuron-wise SNAP (n-SNAP) often outperforms synapse-wise SNAP (s-SNAP) in mitigating catastrophic forgetting, the underlying reasons for this difference are not sufficiently analyzed or explained.


**Minor comments:**
* **The current paper require some structure changes:** Focus more on results in the main paper instead focus on related work and methodology. These sections clearly moved to Appendix. Because the current paper lacks many important details in the main paper text, including the ANN architecture used, training protocol, error analysis, ablation studies, and a clear discussion of implications.

---

### Review · Reviewer_Tj83 · 2026-02-06

**Summary Of Contributions:**

Summary of Contributions

This paper introduces SNAP (Sigmoidal Neuronal Adaptive Plasticity), a biologically inspired weight update modification that aims to mitigate catastrophic forgetting in continual learning. The core idea is to replace standard linear weight growth with sigmoidal weight growth, so that weights which have reached sufficiently large magnitudes become resistant to further change — mimicking Long-Term Potentiation (LTP) in biological synapses. The authors propose two variants: synapse-wise (s-SNAP), where plasticity depends on each individual weight's magnitude, and neuron-wise (n-SNAP), where plasticity depends on the L2 norm of all incoming weights to a neuron. Experiments are conducted on MNIST and FashionMNIST in both i.i.d. and sequential task settings using Hebbian learning and SGD. The main finding is that sigmoidal weight growth prevents catastrophic forgetting in Hebbian learning networks but not in SGD-trained networks.

Key Strengths:

* The biological motivation is clear and intuitive — the connection between LTP consolidation and sigmoidal saturation of weight updates is a natural one, and the paper explains it well.
* The method is simple: it requires no replay buffers, no importance estimation, and no task-specific parameters, avoiding the typical drawbacks of existing continual learning approaches.
* The paper clearly distinguishes between synapse-wise and neuron-wise variants, providing a clean taxonomy (Table 2) of weight growth behaviors.
* The derivation connecting sigmoidal weight growth to a multiplicative plasticity factor (Eq. 12–14) is straightforward and easy to follow.

Key Weaknesses:

* The experimental evaluation is narrow — only MNIST and FashionMNIST with a single-hidden-layer MLP. These are very simple benchmarks, and the paper acknowledges this limitation but does not address it.
* SNAP does not work for SGD-trained networks, which significantly limits its practical applicability given that SGD (and its variants) dominate modern deep learning. The explanation offered (weights don't grow large enough before achieving high accuracy) is somewhat hand-wavy and not rigorously analyzed.
* The sequential task setup (5 binary classification tasks with a shared 10-class output head) is a relatively easy continual learning benchmark. Standard continual learning benchmarks like Split-CIFAR or Permuted MNIST with many more tasks would strengthen the claims.
* There is no comparison against any existing continual learning baseline (e.g., EWC, SI, PackNet), making it hard to contextualize the results.
* The claim of "solving" catastrophic forgetting in Hebbian learning is strong given the limited experimental scope — the forgetting is reduced but performance on earlier tasks still fluctuates noticeably in several figures (e.g., Fig. 4a, 7a-b).

**Audience:**

Yes

**Audience Explanation:**

* **Relevance to continual learning.** Catastrophic forgetting remains one of the central open problems in deep learning. Any novel mechanism that demonstrably reduces forgetting — even in a restricted setting — is of potential interest.
* **Biological plausibility angle** There is growing interest in biologically plausible learning algorithms as alternatives to back-propagation. SNAP proposes a concrete computational analogue of Long-Term Potentiation, which would appeal to researchers bridging neuroscience and machine learning.
* **Simplicity of the mechanism.** The proposed modification — multiplying weight updates by a sigmoidal plasticity factor depending on current weight magnitude — is extremely simple and requires no auxiliary data structures, replay buffers, or task boundaries at inference time. This makes the idea easy to build upon.
* **Novelty of the finding.** The observation that sigmoidal weight growth prevents catastrophic forgetting in Hebbian networks but not in SGD-trained networks is an interesting dissociation that could yield insights into fundamental differences between Hebbian and gradient-based learning.
* **Caveats.** The limited experimental scope (MNIST/FashionMNIST, single-layer MLPs, no baselines) means the current findings may lack sufficient depth or generality for broad impact. Nevertheless, the core idea and preliminary results are sufficient to interest at least a subset of audience.

**Broader Impact Concerns:**

No broader impact concerns. The paper presents a foundational investigation into weight update mechanisms for continual learning, tested on standard image classification benchmarks. It does not raise ethical issues beyond those general to the field.

**Claims And Evidence:**

No

**Claims Explanation:**

The central claims of the paper are only partially supported by the presented evidence, and several gaps undermine the overall convincingness.

* **Narrow benchmark scope.** The experiments are restricted to MNIST and FashionMNIST using a single-hidden-layer MLP with 64 or 96 hidden units. These are among the simplest classification benchmarks in the literature. The claim that SNAP ``stops catastrophic forgetting in Hebbian Learning'' is therefore only demonstrated in a very limited regime. Without evaluation on more challenging datasets (e.g., CIFAR-100, TinyImageNet) or deeper architectures, it is unclear whether the results generalize beyond toy settings.

* **No baselines from the continual learning literature.** The paper positions SNAP against replay, regularization, and parameter isolation methods (Table 1) but does not compare against any of them experimentally. Without comparisons to methods such as EWC, SI, or even simple experience replay, the reader cannot assess the relative merit of SNAP. The claim that SNAP ``avoids the drawbacks seen in previous methods'' is therefore unsupported by empirical evidence.

* **The solving claim is overstated.** Figures 4a, 4c, 7a, and 7b, which represent the best-performing SNAP configurations still show non-trivial fluctuations in per-task accuracy. For instance, in Figure 7b (FashionMNIST, $\lambda=32$), earlier task accuracies visibly degrade when new tasks are introduced. Describing this as solving or stopping catastrophic forgetting is not well-supported by the evidence shown.

* **Incomplete analysis of the SGD failure mode.** The paper acknowledges that SNAP does not work for SGD but offers only a speculative explanation: that weights reach high classification accuracy before growing large enough to enter the sigmoid's consolidation plateau. This hypothesis is not validated with any quantitative analysis (e.g., histograms of weight magnitudes at convergence, tracking plasticity coefficients $|W_{ij}|(1 - |W_{ij}|)$ over training). Without such evidence, the reader is left without a clear understanding of why SNAP fails for SGD, which is a critical gap given that SGD-based learning dominates the field.

* **Limited statistical rigor.** The results appear to be from single runs without error bars, confidence intervals, or reporting of variance across seeds. Given the sensitivity to hyperparameters $\lambda$, $\eta$, and $\alpha$ (visible in Figures 2 and 3), it is difficult to assess the robustness of the reported results.

* **Hyperparameter sensitivity.** Figures 2 and 3 show that performance varies substantially across values of $\lambda$ for different weight growth configurations. The paper does not adequately discuss how sensitive SNAP is to this hyperparameter in the sequential setting, nor does it provide guidance on how to select $\lambda$ in practice without access to i.i.d.\ validation data.

In summary, while the core idea is interesting and the qualitative trend (sigmoidal growth helps Hebbian networks retain prior task performance) is visible in the figures, the evidence is not sufficiently rigorous, comprehensive, or well-contextualized to convincingly support the paper's central claims.

**Requested Changes:**

Critical:

* **Add continual learning baselines.** Compare against at least EWC or SI on the same benchmarks. Without this, SNAP's relative merit cannot be assessed.
* **Evaluate beyond MNIST/FashionMNIST.** Test on Split-CIFAR-10 or Split-CIFAR-100 with deeper architectures to support generality claims.
* **Report error bars.** Include means and standard deviations over multiple random seeds (≥3). Single-run results are insufficient given the hyperparameter sensitivity shown in Figures 2–3.
* **Analyze the SGD failure quantitatively.** Provide weight magnitude histograms or plasticity coefficient trajectories comparing Hebbian vs. SGD networks. The current explanation is speculative.
* **Temper the "solving/stopping" language.** Figures (especially FashionMNIST) show non-trivial accuracy fluctuations on earlier tasks. "Mitigating" or "significantly reducing" forgetting would be more accurate.

Recommended:

* **Report standard CL metrics** (Average Accuracy, Backward Transfer) rather than relying on visual inspection of curves.
* **Ablate initialization scale** since SNAP's plasticity factor $∣W∣(1−∣W∣)|W|(1-|W|)
∣W∣(1−∣W∣)$ is sensitive to absolute weight magnitude.
* **Test with more sequential tasks** (e.g., 10 tasks) to probe the capacity–plasticity tradeoff as neurons saturate.
* **Clarify whether λ was tuned on sequential performance**
 or transferred from i.i.d. experiments, as the former would be unrealistic in true continual learning.
* **Experiment with alternative saturating functions** (e.g., $\tanh$) to determine whether the benefit is specific to the sigmoid or general to any saturating nonlinearity.

---

> ### Author Response · Authors · 2026-02-10
> **Response to reviewer Tj83 – we generally agree and will implement changes**
>
> We thank the reviewer for this helpful and thoughtful review. We mostly agree with the reviewer's assessment and will implement most of the reviewer's suggestions which we believe will make for a much stronger paper. However we have a few clarifications regarding the current state of Hebbian learning in ANNs.
>
> ---
>
> ## Key Clarifications
>
> **Hebbian learning does not currently work with deeper networks or CIFAR-10/100.** This is a field-wide limitation, not specific to our work. Nearly all recent Hebbian learning papers are limited to MNIST/FashionMNIST and shallow architectures.
>
> **EWC and Synaptic Intelligence are SGD-specific methods.** They gradient-based computations, making them incompatible with Hebbian learning. We will add them as baselines for our SGD experiments.
>
> ---
>
> ## Response to Critical Changes
>
> ### 1. Add Continual Learning Baselines
>
> **For SGD:** We will add EWC and SI baselines to our SGD experiments.
>
> **For Hebbian:** EWC and SI cannot be applied as they are gradient-based methods. We note that both also require knowledge of task boundaries, which SNAP does not. To the best of our knowledge, our method is the first continual learning method for discrete time Hebbian Learning.
>
> ### 2. Evaluate Beyond MNIST/FashionMNIST
>
> **CIFAR-10/100 and deeper architectures** are not currently feasible with Hebbian learning.
>
> **We will add:** Permuted MNIST experiments as suggested by the reviewer, testing with 10-20 tasks.
>
> ### 3. Report Error Bars
>
> Agreed. We will add error bars.
>
> ### 4. Analyze the SGD Failure Quantitatively
>
> Agreed. We will add:
>
> - Weight magnitude histograms comparing Hebbian vs. SGD
> - Plasticity coefficient trajectories over training
> - Quantitative analysis of weight saturation rates
>
> ### 5. Temper the "Solving/Stopping" Language
>
> Agreed. We will change the title to "Mitigating" and use more measured language throughout (e.g., "significantly reduces catastrophic forgetting").
>
> ---
>
> ## Response to Recommended Changes
>
> ### 1. Report Standard CL Metrics
>
> Agreed. We will report Average Accuracy and Backward Transfer for all experiments.
>
> ### 2. Ablate Initialization Scale
>
> Agreed. We will test different initialization scales and analyze their effect on SNAP's performance.
>
> ### 3. Test with More Sequential Tasks
>
> Agreed. We will extend experiments to 10-20 tasks using Permuted MNIST and potentially Extended MNIST/Omniglot.
>
> ### 4. Clarify λ Tuning
>
> We will clarify our hyperparameter selection protocol, λ was selected in the i.i.d setting.
>
> ### 5. Experiment with Alternative Saturating Functions
>
> Great idea. We will test tanh-based weight growth and other saturating functions to determine if benefits are general to saturating nonlinearities.

---

### Review · Reviewer_sKqh · 2026-03-29

**Summary Of Contributions:**

The authors introduce Sigmoidal Neuronal Adaptive Plasticity (SNAP), a novel weight growth algorithm designed to mitigate catastrophic forgetting in Artificial Neural Networks (ANNs). Inspired by the biological process of Long-Term Potentiation (LTP), SNAP replaces standard linear weight updates of back propagation with a signoidal growth. This allows weights to learn rapidly during an intermediate phase but stabilize once they reach extreme values making them resistant to further changes. The authors evaluate SNAP on sequential task learning using both Hebbian and Stochastic Gradient Descent (SGD) learning. The key finding is that sigmoidal weight growth prevents catastrophic forgetting in Hebbian networks without relying on data replay or parameter isolation mechnism.

**Audience:**

Yes

**Audience Explanation:**

The findings will be of significant interest to researchers in the fields of biological machine learning, neuromorphic computing, and continual learning.  Especially, this paper provide novel insights in continual learning in that this mitigates catastrophic forgetting  by modifying pure update rules via Hebbian learning without relying on data-replay or auxiliary learnable weights.

**Broader Impact Concerns:**

There are no obvious or immediate ethical concerns.

**Claims And Evidence:**

Yes

**Claims Explanation:**

The theoretical derivation of SNAP is quite rigorous. The authors clearly demonstrate how to modify a standard linear update rule to sigmoidal update rule. Furthermore, experiments of sequential task learning provides a standard baseline for evaluating continual learning. The ablation studies systematically comparing linear, exponential, and sigmoidal weight growth across different network layers (hidden layer and classification layer) provide a comprehensive understanding of the proposed algorithms's mechanism.

**Requested Changes:**

Major

- **Discuss FashionMNIST Results in the Main Text**: The performance degradation on FashionMNIST (Appendix A.8) is a critical limitation. The authors should move this discussion into the main text to transparently address the method's boundaries and explain the performance gap compared to MNIST.
- **Evaluate on Deeper Architectures**: The current experiments are limited to a shallow MLP. Validating SNAP across diverse, deeper network topologies (e.g., CNNs or small Vision Transformers) would significantly strengthen the paper's impact.
- **Test on Complex Datasets**: As acknowledged in the Limitations section, the paper lacks evidence that this method scales to complex, high-dimensional data. Extending the sequential learning experiments to datasets like CIFAR-10 or CIFAR-100 would provide much stronger evidence for the method's broad applicability.

Minor

- **Soften the Title's Claims**: The title's claim of "Stopping Catastrophic Forgetting"  is overstated. Given the reliance on a single-hidden-layer MLP and the noticeable forgetting observed in the FashionMNIST experiments (Figures 7b, 7c), the authors should use more precise terms like "mitigating" or "reducing."

---

> ### Author Response · Authors · 2026-04-02
> **Response to Reviewer sKqh -- Thank you**
>
> Thank you for the fair and incisive review that will improve our paper.
>
> We would first like to note that Hebbian learning does not currently work with deeper networks or CIFAR-10/100. This is a field-wide limitation, not specific to our work. Nearly all recent Hebbian learning papers are limited to MNIST/FashionMNIST and shallow architectures. It is our fault for not having made that clear in the paper. This is something which we will correct in the revised version.
>
> As for the other major request change "Discuss FashionMNIST Results in the Main Text", we agree with the reviewer and will do that.
>
> We also agree with the reviewer about softening the title's claim and will use "Mitigating" rather than "Stopping" as recommended by the reviewer.

---

### Decision · Action_Editor_dMaF · 2026-05-20

**Recommendation:** Reject

**Audience:**

Yes

**Audience Explanation:**

Hebbian learning has attracted interest in the neural network community because it offers a simple yet biologically plausible learning mechanism. Since this work presents an interesting and novel observation that sigmoidal weight growth in Hebbian networks can naturally mitigate catastrophic forgetting, researchers interested in biologically inspired continual learning may find the paper’s findings valuable.

**Claims And Evidence:**

No

**Claims Explanation:**

Two out of the three reviewers still feel that the answer to this question is negative, as the authors’ responses do not fully address their initial concerns. Reviewer Qz3z noted that the rebuttal did not sufficiently resolve issues regarding the limited evaluation and discussion of the results, the lack of hyperparameter analysis, and the incomplete related work discussion with missing citations. Reviewer Tj83 raised a similar set of concerns and felt that, although the authors promised improvements, these revisions have not yet been demonstrated in the manuscript.

**Resubmission Of Major Revision:**

The authors may consider submitting a major revision at a later time.